# Univariate and multivariate genomic prediction for agronomic traits in durum wheat under two field conditions

**Paolo Vitale**[1,2,3]*, **Giovanni Laidò**[1], **Gabriella Dono**[1,4], **Ivano Pecorella**[1],
**Vishnu Ramasubramanian**[5], **Aaron Lorenz**[5], **Pasquale De Vita**[1], **Nicola Pecchioni**[1]

1 Research Centre for Cereal and Industrial Crops (CREA-CI), CREA—Council for Agricultural Research and Economics, Foggia, Italy, 2 Department of Agriculture, Food, Natural Science, Engineering, University of Foggia, Foggia, Italy, 3 International Maize and Wheat Improvement Center (CIMMYT), Texcoco, Edo, de México, México, 4 Charles Darwin Department of Biology and Biotechnologies, University of Rome, Rome, Italy, 5 Department of Agronomy and Plant Genetics, University of Minnesota, Saint Paul, MN, United States of America

* p.vitale@cgiar.org

**Data Availability Statement:** Molecular markers were uploaded in the public repository figshare (https://doi.org/10.6084/m9.figshare.22674667. v1). Similarly, the phenotypic dataset can be

## Abstract

Genomic prediction (GP) has been evaluated in durum wheat breeding programs for several years, but prediction accuracy (PA) remains insufficient for some traits. Recently, multivariate (MV) analysis has gained much attention due to its potential to significantly improve PA. In this study, PA was evaluated for several agronomic traits using a univariate (UV) model in durum wheat, subsequently, different multivariate genomic prediction models were performed to attempt to increase PA. The panel was phenotyped for 10 agronomic traits over two consecutive crop seasons and under two different field conditions: high nitrogen and well-watered (HNW), and low nitrogen and rainfed (LNR). Multivariate GP was implemented using two cross-validation (CV) schemes: MV-CV1, testing the model for each target trait using only the markers, and MV-CV2, testing the model for each target trait using additional phenotypic information. These two MV-CVs were applied in two different analyses: modelling the same trait under both HNW and LNR conditions, and modelling grain yield together with the five most genetically correlated traits. PA for all traits in HNW was higher than LNR for the same trait, except for the trait yellow index. Among all traits, PA ranged from 0.34 (NDVI in LNR) to 0.74 (test weight in HNW). In modelling the same traits in both HNW and LNR, MV-CV1 produced improvements in PA up to 12.45% (NDVI in LNR) compared to the univariate model. By contrast, MV-CV2 increased PA up to 56.72% (thousand kernel weight in LNR). The MV-CV1 scheme did not improve PA for grain yield when it was modelled with the five most genetically correlated traits, whereas MV-CV2 significantly improved PA by up to ~18%. This study demonstrated that increases in prediction accuracy for agronomic traits can be achieved by modelling the same traits in two different field conditions using MV-CV2. In addition, the effectiveness of MV-CV2 was established when grain yield was modelled with additional correlated traits.

obtained by downloading from the public repository Figshare (10.6084/m9.figshare. 25554993).

**Funding:** The research was partially funded by European Commission, Grant Agreement number: 727247 — SolACE — H2020-SFS-2016-2017/ H2020-SFS-2016-2, and by Italian Ministry of Agriculture, Food and Forestry Policies (MiPAAF), sub-project 'Tecnologie digitali integrate per il rafforzamento sostenibile di produzioni e trasformazioni agroalimentari (AgroFiliere)' (AgriDigit programme) (DM 36503.7305.2018 of 20/12/2018). No additional external funding was received for this study.

**Competing interests:** The authors have declared that no competing interests exist.

## Introduction

Durum wheat (*Triticum turgidum* L. ssp. *durum* Desf.) plays a crucial role in global agriculture, accounting for approximately 5% of the total wheat crop, with an annual production of 40 million tonnes [1]. It is mainly cultivated in the Mediterranean basin, an area characterized by strong climatic instability that limits yield and grain quality. This region is particularly vulnerable to climate change effects such as drought and salinity [2]. The new climate scenario potentially undermines past efforts to improve yield and meet global food security goals [3]. In recent years, advancements in genotyping technologies, especially the identification of Single Nucleotide Polymorphisms (SNPs), have revolutionized breeding applications [4]. Molecular breeding techniques leverage these advancements to enhance genetic gain, offering advantages such as cost-effective genotyping and the ability to make selections during off-seasons, ultimately saving time [5].

Genomic selection (GS) becomes an attractive plant breeding method for durum wheat and other many species, increasing the genetic gain per unit of time [6, 7]. Compared to Marker-Assisted Selection (MAS), GS is particularly advantageous for traits controlled by multiple genes, enabling the capture of both major and minor Quantitative Trait Loci (QTL) [8]. Therefore, GS appears more attractive than MAS for selecting complex agronomic traits, which generally have low heritability values and whose expression is often masked by climatic trends and characteristics of the cultivation environment [9]. In the GS method, a statistical model is trained using genotypic and phenotypic information from a set of individuals known as the training population (TP). The model is then used to calculate the genomic estimated breeding values (GEBVs) of a set of genotypes named the validation population (VP). The correlation between GEBVs and true breeding values (TBVs) determines to the genomic prediction accuracy (PA) of the model [10, 11]. Subsequently, the trained model is applied to a breeding population (BP), which has only been genotyped, to select new genotypes based on their GEBVs [12]. Univariate (UV) genomic prediction, which computes a single trait at a time, has been commonly utilized in durum wheat breeding to enhance various characteristics, including grain yield, quality traits [13–17], phenological traits [18, 19], biotic [20–22], and abiotic stress [23].

For durum wheat, as well as for all crops, the ultimate goal of breeding is to increase genetic gain. Genetic gain per year can be estimated as $\Delta G = i\, r\, \sigma_A/t$ where $\Delta G$ is the response to selection, $i$ corresponds to the selection intensity, $r$ is the selection accuracy, $\sigma_A$ is the square root of the additive genetic variance, and $t$ is the duration of the breeding cycle [5]. In a GS-based breeding program, $r$ corresponds to the correlation between TBVs and GEBVs, or the genomic prediction accuracy. Prediction accuracy is a complex parameter to discern since it is influenced by several variables, such as population size [24] marker density [25], heritability [10], and the genetic architecture of the trait of interest [26], the relatedness among individuals in the TP and VP [27] and the prediction model [28].

Therefore, improving genomic prediction accuracy is a crucial goal for many breeders, statisticians, and quantitative geneticists [26, 29]. Consequently, multivariate (MV) genomic prediction, which considers multiple variables simultaneously, is gaining widespread popularity. This approach allows for the simultaneous exploitation of information from multiple traits, thereby enhancing the genomic prediction accuracy [30]. Several studies in the literature have already highlighted the effectiveness of implementing the multivariate approach to increase prediction accuracy for a target trait due to borrowing information from secondary traits. For instance, Jia and Jannink [30] compared uni- and multivariate GS models and reported that prediction accuracy significantly increased for low-heritability traits when these were modelled alongside correlated high-heritability traits. Lado and colleagues [29] evaluated the prediction

accuracy using uni- and multivariate models and found the supremacy of an MV model when one highly correlated secondary trait is phenotyped in both the TP and VP. In addition, prediction accuracy for grain yield was significantly increased by the multivariate approach and, in some cases, it was improved by ~100% compared to the single-trait method in spring wheat [31]. The multivariate GP approach has also been successful in enhancing the prediction accuracy of deoxynivalenol content across various populations of soft red winter wheat [32]. Additionally, the integration of near-infrared (NIR) technology information into GP models aimed to enhance PA of diverse end-product quality traits in wheat. Notably, the authors observed a substantial 30% increase in PA by incorporating NIR-predicted data, surpassing the results obtained from a single-trait analysis [33]. A recent study examined various agronomic traits, including grain yield, thousand kernel weight, across 237 bread wheat lines. The comparison between single-trait and multi-trait models revealed the superiority of the latter, demonstrating a genetic gain ranging from 5% to 22% more than the single-trait models [34]. Similarly, multi-trait genomic selection proved to be more effective than single-trait models for predicting micronutrients such as zinc and iron in wheat breeding [35]. Kaushal et al. [36] evaluated high-throughput phenotyping-based (HTP) traits in MV-GBLUP (genomic best linear unbiased predictors) models for predicting grain yield, test weight, and grain protein content. While UV-GBLUP achieved a PA of 0.23 for grain yield, MV-GBLUP, incorporating HTP traits collected during the booting stage, significantly improved PA, with an increase of 60 Furthermore, multi-trait models that combine direct measurements of end-product quality traits with their NIR predictions achieved higher prediction accuracy than single-trait models [33]. It was also demonstrated that the Fusarium-damaged kernels trait, assessed through a neural network and cell phone camera images, can enhance genomic prediction accuracy for deoxynivalenol when used as a secondary trait in a multi-trait GS model [37]. However, a study evaluating genomic prediction accuracy for three wheat diseases (tan spot, spot blotch, and septoria nodorum blotch) found no advantage in modeling all diseases together compared to using a single-trait approach in a synthetic hexaploid wheat population [38].

In this study, several multivariate genomic prediction models were implemented for various traits and indices related to grain yield, quality, and crop phenology, observed in two different field conditions in a panel of 250 durum wheat genotypes using two cross-validation schemes (CV1 and CV2). Our aims were (i) to compare the accuracy of uni- and multivariate genomic prediction models for several agronomic traits under two field conditions, and (ii) to test whether the prediction accuracy of grain yield can be improved by including the most genetically correlated traits in MV genomic prediction models. Results from this study will illuminate the usefulness of MV genomic prediction models for durum wheat breeding as well as for similar breeding programs.

## Materials and methods

### Plant material and field trials

The two hundred and fifty durum wheat varieties used in this study constituted the SolACE project durum wheat association panel (https://www.solace-eu.net/). The panel included 200 elite genotypes derived from Italian, French, and American breeding programs, as well as 50 inbred lines developed by the French National Research Institute for Agriculture, Food and the Environment (INRAE) from an advanced Evolutionary Pre-breeding Population (EPO). The field trial was conducted over two consecutive growing seasons (2017–2018 and 2018–2019) at the experimental farm of the Research Centre for Cereal and Industrial Crops at Foggia, south Italy (CREA-CI) (41˚27'40.2" N 15˚30'04.5" E). The genotypes were grown on clay soil (United States Department of Agriculture Classification, Washington, DC, USA) with the

following main agrochemical characteristics: organic matter (Walkley-Black method) 2.5 and 2.6% in 2018 and 2019, respectively; available phosphorus (Olsen method) 62.0 and 68.0 mg kg$^{-1}$; exchangeable potassium (ammonium acetate method) 422 and 450 mg kg$^{-1}$; total nitrogen (Dumas method) 1.3 and 1.1%. The fields were homogeneous and without preceding crops ("set-aside land"). The sowing was carried out on November 28$^{th}$ and December 7$^{th}$ in 2017 and 2018 respectively. An ordinary sowing density of approximately 350 seeds m$^{-2}$ was used. The plot dimensions were 2.4 m$^2$ and 4.8 m$^2$ for the first and second years of the trial, respectively. Plots were arranged in a randomized complete block design with two replications. The genotypes were randomly sown and arranged in a split-plot design, with two agronomic treatments assigned to the main plots. The treatments included I) irrigation water supply and high nitrogen input, referred to as HNW, and II) rainfed conditions and low nitrogen input, referred to as LNR. In the high input treatment, 240 kg/ha of N fertilizer was divided into three applications (120, 70, and 50 N kg/ha at tillering, stem elongation, and flowering time respectively), and the plots were irrigated using a drip irrigation system maintaining soil moisture not less than 20% of field capacity. Soil moisture probes evenly distributed across the fields provided data to help ensure target field capacity was maintained. The low input treatment was carried out under rainfed conditions and with a single application of 60 kg/ha of nitrogen in the tillering phase. Durum wheat grains were machine-harvested at full maturity on the 25$^{th}$ and 27$^{th}$ of June in 2018 and 2019, respectively, using a Wintersteiger Nursery Master Elite plot combine (Wintersteiger Inc., Ried im Innkreis, Austria). Finally, the total precipitations were also recorded, expressed in mm for each crop season using the public online software NASA POWER (https://power.larc.nasa.gov, available online on 29 November 2021). Taking into account the water needs of durum wheat, precipitations were observed from June to the following June of each crop season to better investigate the field management effect.

## Trait phenotyping

During the two growing seasons, wheat plants were phenotyped for ten agronomic traits in both HNW and LNR management conditions. The plants were monitored twice a week to record the date of flag leaf appearance (FLA), days to heading (DTHD), days to anthesis (DTA), and days to maturity (DTM) of the main stem (i.e., growth stages 47, 55, 65 and 90) [39] expressed as days from 1$^{st}$ April. The normalized difference vegetation index (NDVI) was recorded at anthesis on each plot by using the GreenSeeker™ Trimble Inc. handheld optical sensor unit. At harvest, grain yield (GY) (t/ha) was assessed for each plot at 15% moisture content. Grain protein content (GPC) (%), test weight (TW) (kg hl$^{-1}$), and semolina yellow index (YI) (ppm) were determined by near-infrared reflectance spectroscopy using an Infratec 1229 Grain Analyzer (Foss Tecator, Hillerød, Denmark). Thousand kernel weight (TKW) (g) was measured as the mean weight of three sets of 500 grains per plot.

By using observed traits, two derivative parameters were calculated to better characterize the entire durum wheat panel. Grain protein deviation (GPD) was expressed as the deviation from the regression between GY and GPC. This was achieved by calculating the residuals from the regression of protein content on grain yield based on environmental means [40] using the following formula:

$$GPD = GPC - \alpha - \beta GY \tag{1}$$

Where GPC is the observed values for protein content, GY is the observed grain yield values, and $\alpha$ and $\beta$ are estimates of the intercept and coefficient of the regression between GPC and GY, respectively. Changing the role of protein content and grain yield in the equation above brings forth grain yield deviations (GYD) [15] that were also considered as a viable selection

criterion:

$$GYD = GY - \alpha - \beta GPC \tag{2}$$

## E-BLUEs, heritability, and genetic correlations

Empirical best linear unbiased estimates (E-BLUEs) were calculated from the observed data based on the random complete block design. The E-BLUE values were estimated using a restricted maximum likelihood (REML) approach by using the following linear mixed model:

$$y_{ijk} = \mu + g_i + e_j + r_{k(j)} + ge_{ij} + \varepsilon_{ijk} \tag{3}$$

where $y_{ijk}$ is the observed value, $\mu$ is the overall mean, $g_i$ the effect of the $i$th line assumed as a fixed effect, $e_j$ is the $j$th environment (year) effect modelled as the random effect, $r_{k(j)}$ is the random effect of the $k$th replicate in the $j$th environment, $ge_{ij}$ are the genotype by environment interaction, and $\varepsilon_{ijk}$ corresponded to the residual effect considered a random effect and assumed to have a normal distribution $\varepsilon_{ijk} \sim N(0, \sigma_\varepsilon^2)$. This model was fit using the *remlf90* function of the "breedR" package in R [41]. Estimates of broad-sense heritability ($H^2$) were obtained using Eq 3 for the calculation of E-BLUEs except for fitting $g_i$ as a random effect to estimate the genotypic variance. The following formula was used to calculate the broad-sense heritability:

$$H^2 = \frac{\sigma_g^2}{\sigma_g^2 + \frac{\sigma_{ge}^2}{y} + \frac{\sigma_\varepsilon^2}{yr}} \tag{4}$$

where $\sigma_g^2$ is the genotypic variance, $\sigma_{ge}^2$ is the variance coming from the genotype by environment (year) interaction, y is the number of years, r is the number of replications and $\sigma_\varepsilon^2$ is the error variance [42]. Finally, genetic correlations which measure the extent to which the variation in two traits is due to shared genetic factors were calculated. These correlations provide an estimate of the additive genetic effect shared between two traits. Genetic correlations between the two traits were calculated as follows:

$$\rho_{g_{ij}} = \frac{\sigma_{g_{ij}}}{\sigma_{g_i} \sigma_{g_j}} \tag{5}$$

Where $\sigma_{g_{ij}}$ is the genetic covariance between trait $i$ and $j$, $\sigma_{g_i}$ and $\sigma_{g_j}$ represent the square root of the genetic variances of $i$ and $j$ traits respectively. Genetic variance and covariance were estimated using both phenotypic and genotypic information through the function *multitrait* in the R package "BGLR" [43].

## Genomic prediction models

Before harvest, leaf tissue of each genotype was collected at the third-leaf stage and stored at -80°C. DNA was extracted from the stored tissue using a Sbeadex Livestock kit (LGC Genomics GmbH, Germany). Subsequently, genotyping of high-quality DNA was performed using a 420K SNP Axiom Array at Gentyane, France (INRA, Clermont-Ferrand, France: http://gentyane.clermont.inra.fr/). Raw data were filtered using a minor allele frequency (MAF) cut-off of <5%, and missing data cut-off of <20%. The conversion from the Hapmap format to a numerical matrix was conducted using the software Tassel 5.0 (https://tassel.bitbucket.io). The software assigns a value of 1 to represent homozygous major allele, 0 for homozygous minor

allele, and 0.5 for heterozygous genotypes. To align the numerical matrix with the used tools, a substitution code was applied, replacing the values with -1, 1, and 0 respectively.

E-BLUEs were used to perform all genomic prediction analyses in both uni- and multivariate approaches. The univariate genomic best linear unbiased prediction (UV-GBLUP) model was utilized as a reference for all multivariate models, which included MV-GBLUP, Bayesian ridge regression (BRR), Bayesian reproducing kernel Hilbert spaces regressions (RKHS), spike and slab regression (SpikeSlab), and machine learning random forest (RF). In particular, the univariate GBLUP model was formulated as the following equation:

$$\mathbf{y} = \mu + \mathbf{Zg} + \boldsymbol{\varepsilon} \tag{6}$$

where $\mathbf{y}$ is the vector of the E-BLUEs, $\mu$ is the grand mean, $\mathbf{Z}$ is the design matrix of random effects, $\mathbf{g}$ is the vector of genomic breeding values, and $\boldsymbol{\varepsilon}$ is the vector of random residuals. In this model, it assumed that $\mathbf{g} \sim N(0, \mathbf{G}\sigma_G^2)$, where $\mathbf{G}$ is the genomic relationship matrix built from the SNP matrix, and $\sigma_G^2$ is the additive genetic variance [44]. By contrast, multivariate GBLUP was performed using the following model:

$$\begin{bmatrix} \mathbf{y}_1 \\ \vdots \\ \mathbf{y}_P \end{bmatrix} = \begin{bmatrix} \mathbf{X}_1 & \cdots & 0 \\ \vdots & \ddots & \vdots \\ 0 & \cdots & \mathbf{X}_P \end{bmatrix} \begin{bmatrix} \mu_1 \\ \vdots \\ \mu_P \end{bmatrix} + \begin{bmatrix} \mathbf{Z}_1 & \cdots & 0 \\ \vdots & \ddots & \vdots \\ 0 & \cdots & \mathbf{Z}_P \end{bmatrix} \begin{bmatrix} \mathbf{g}_1 \\ \vdots \\ \mathbf{g}_P \end{bmatrix} + \begin{bmatrix} \boldsymbol{\varepsilon}_1 \\ \vdots \\ \boldsymbol{\varepsilon}_P \end{bmatrix} \tag{7}$$

Where $\mathbf{y}$ is the vector of BLUEs of p traits, $\mathbf{X}$ and $\mathbf{Z}$ are the design matrices for fixed and random effects, $\begin{bmatrix} \mu_1 \\ \vdots \\ \mu_P \end{bmatrix}$ correspond to trait intercepts of p traits, the predicted genetic values,

denoted as $\begin{bmatrix} \mathbf{g}_1 \\ \vdots \\ \mathbf{g}_P \end{bmatrix}$, were assumed to distributed as $\begin{bmatrix} \mathbf{g}_1 \\ \vdots \\ \mathbf{g}_P \end{bmatrix} \sim N(0, \mathbf{H} \otimes \mathbf{G})$, where $\mathbf{G}$ is the geno-

mic relationship matrix estimated from the markers, $\mathbf{H}$ is the unstructured variance-covariance matrix for the genetic effects among traits, and $\otimes$ is the Kronecker product. It was also

assumed that the residual term, $\begin{bmatrix} \boldsymbol{\varepsilon}_1 \\ \vdots \\ \boldsymbol{\varepsilon}_P \end{bmatrix}$, followed a distribution $\begin{bmatrix} \boldsymbol{\varepsilon}_1 \\ \vdots \\ \boldsymbol{\varepsilon}_P \end{bmatrix} \sim N(0, \mathbf{R} \otimes \mathbf{I})$, where $\mathbf{R}$

is the diagonal variance-covariance matrix for the residuals effect among the traits, and $\mathbf{I}$ is the identity matrix. This model was implemented for all traits, including the same trait in both HNW and LNR conditions. Subsequently, GY was used as the primary trait, and the five most genetically correlated traits were used as secondary traits, keeping the field conditions separate. Both UV- and MV-GBLUP were trained using the *mmer* function of the "sommer" package [45]. Two SNP-based models such as BRR, and SpikeSlab were applied by using the *multitrait* function implemented in the "BGLR" R package [43] with the following data equation:

$$\mathbf{Y} = \mathbf{1}\boldsymbol{\mu}' + \mathbf{X}_1 \mathbf{B}_1 + \mathbf{E} \tag{8}$$

where $\mathbf{Y}$ represents the matrix of the E-BLUEs for each individual and for each trait, $\mu = (\mu_1, \ldots, \mu_t)'$ are the trait-specific intercepts, $\mathbf{X}_1$ corresponds to the incidence matrix of the set of the predictors (the molecular markers), $\mathbf{B}_1$ represents the matrix of effects, and $\mathbf{E}$ is the error

matrix assumed to be independent and identically distributed (IID). In the error matrix, the rows correspond to the individuals, and each row follows a multivariate normal distribution (MVN) with zero mean and covariance matrix $\mathbf{R}_0$. Hence, the conditional distribution of the data, given the regression parameters and the error covariance matrix, is as follows:

$$p(\mathbf{Y}|\boldsymbol{\theta}) = \prod_{i=1}^{n} MVM(\boldsymbol{y}_i|\boldsymbol{\eta}_i, \mathbf{R}_0) \tag{9}$$

where $\boldsymbol{\eta}_i = \mu + \mathbf{B}'_1\mathbf{x}_{1i}$, which is a vector containing the conditional mean of the $i$th observations for the $t$th trait where $\mathbf{x}_{1i}$ correspond to the $i$th rows referring to the matrix $\mathbf{X}_1$, and $\boldsymbol{\theta} = \{\mu, \mathbf{B}_1, \mathbf{R}_0\}$. In addition, an individual-based model (RKHS) was also performed by applying the *multitrait* function with a modified data equation:

$$\mathbf{Y} = \mathbf{1}\boldsymbol{\mu}' + \mathbf{U}_1 + \mathbf{U}_2 + \mathbf{U}_3 + \mathbf{E} \tag{10}$$

Where $\mathbf{Y}, \boldsymbol{\mu},$ and $\mathbf{E}$ maintain their respective meaning from Eq (8). In addition, $\mathbf{U}_1, \mathbf{U}_2,$ and $\mathbf{U}_3$ are the matrices of random effects which in our study correspond to the three kernel relationship matrices $\mathbf{K}_*$ derived from the markers [43, 46]. In Eq (10), the conditional distribution of the data, given the regression parameters and the error covariance matrix, can be described with Eq (9) where $\boldsymbol{\eta}_i = \mu + \mathbf{u}_{1i} + \mathbf{u}_{2i} + \mathbf{u}_{3i}$, where $\mathbf{u}_{*i}$ is the $i$th rows corresponding to the $\mathbf{U}_*$ matrices, and $\theta = \{\mu, \mathbf{U}_1, \mathbf{U}_2, \mathbf{U}_3, \mathbf{R}_0\}$.

Finally, a multivariate machine learning random forest model was applied. Bootstrap samples were taken from the training population set and a random forest tree ($T_B$) was subsequently built using the bootstrapped data with a splitting criterion which was specific for each variable. The predicted values for the validation population were calculated using the following equation:

$$\hat{y}_i = \frac{1}{B}\sum_{b=1}^{B} T_b(x_i) \tag{11}$$

where $\hat{y}_i$ is the predicted value for the $i$th individuals with genotype $x_i$, $B$ is the number of bootstrap samples, and $T$ is the number of trees. The hyperparameters optimization was performed before fitting the model using an inner grid search cross-validation function. Different sets of hyperparameters were considered, including the number of trees (100, 200, 300), the number of variables to possibly split at each node (80, 100, 120), and the minimum size of the terminal node (3, 6, 9). The latter model was run using the R function *rfsrc* which is implemented in the package "randomForestSRC" [47]. The random forest model was complete only in the MT-CV1 scheme because of the inability of the *rfsrc* R function to digest NA values. In this study, prediction accuracy was estimated using the correlation between the genomic estimated breeding values and the E-BLUEs. All the code for performing genomic prediction analyses is available in the following GitHub repository: https://github.com/paolovitale777/Code-univariate-and-multivariate-genomic-prediction/blob/main/Code. Finally, a mean separation test (Duncan Test) was performed to compare sets of means and discern statistical differences among prediction accuracies from all uni- and multivariate models. Duncan Test was applied using the function *duncan.test* implemented in the "agricolae" R package (version 1.4.0) [48].

## Cross-validation schemes

Prediction accuracy was assessed through five-fold cross-validation, repeated 10 times, using two distinct schemes (CV1 and CV2) for two types of analysis. In the univariate approach, only cross-validation 1 (UV-CV1) was used, where 4/5 of the population served as the training population (TP), and the remaining 1/5 was designated as the validation population (VP)

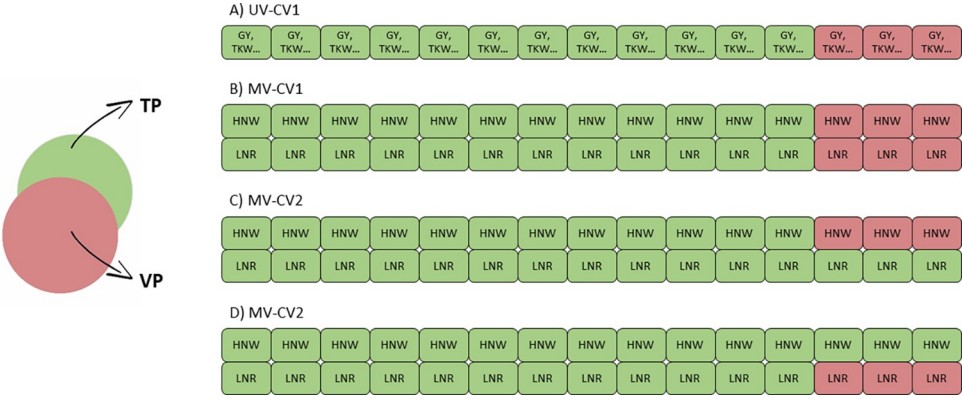

**Fig 1. Cross-validation scheme:** *Same trait in two field conditions.* The training population (TP) is represented in green, and the validation population (VP) in red. **(A)** Univariate cross-validation 1 (UV-CV1): this scheme was applied for all traits, with 4/5 of the population used for training the model and 1/5 for validation. **(B)** Multivariate cross-validation 1 (MV-CV1): the phenotypical information for the same trait in both high nitrogen and well-watered (HNW) and low nitrogen and rainfed (LNR) conditions were included in the model for the individuals involved in the TP. The correlation between the genomic estimated breeding values (GEBVs) and the BLUEs was performed separately for the target trait in its HNW and LNR conditions. **(C)** Multivariate cross-validation 2 (MV-CV2): phenotypic information from the LNR condition of the individuals in the VP was included in the model to predict the HNW condition. **(D)** Multivariate cross-validation 2 (MV-CV2): the same scenario as in **(C)**, but HNW and LNR conditions reversed.

(Fig 1A). In the multivariate approach, two analyses were conducted: one encompassing all traits and another specific to GY, using both CV1 and CV2.

## Same trait in two field conditions

Data from each trait across both field conditions were incorporated into the model (Fig 1). Two schemes were used for analysis: multivariate cross-validation 1 (MV-CV1) and multivariate cross-validation 2 (MV-CV2). In MV-CV1, a randomly selected 4/5 subset of the population was used to train a model incorporating genotypic and phenotypic data from both field conditions, predicting the remaining 1/5 genotypes for each field condition separately (Fig 1B). In MV-CV2, the model was trained using genotypic information and phenotypic data from the HNW condition for 4/5 of the population and phenotypic data from the LNR condition for all individuals. This model was used to predict the remaining genotypes in HNW conditions (Fig 1C). The reversed scenario was also shown in Fig 1D (as in Lado et al. [29]).

## Grain yield with correlated traits

In the second analysis, focusing on grain yield and correlated traits, prediction accuracy was calculated by including the five traits most genetically correlated to GY (Fig 2). For the HNW prediction, the predictive model included TKW, TW, GPC, YI, and FLA, while for LNR prediction, TKW, TW, NDVI, GPC, and YI were used. This analysis was performed separately for HNW and LNR conditions. In MV-CV1, a random 4/5 subset was sampled to train the model using genotypic and phenotypic information of GY and the correlated traits. The model was then validated on the remaining individuals for GY (Fig 2B). In MV-CV2, the model was trained using GY for a random 4/5 subset, supplemented with phenotypic information of other traits for the entire panel (Fig 2C), predicting GEBVs for 1/5 of the remaining genotypes for GY.

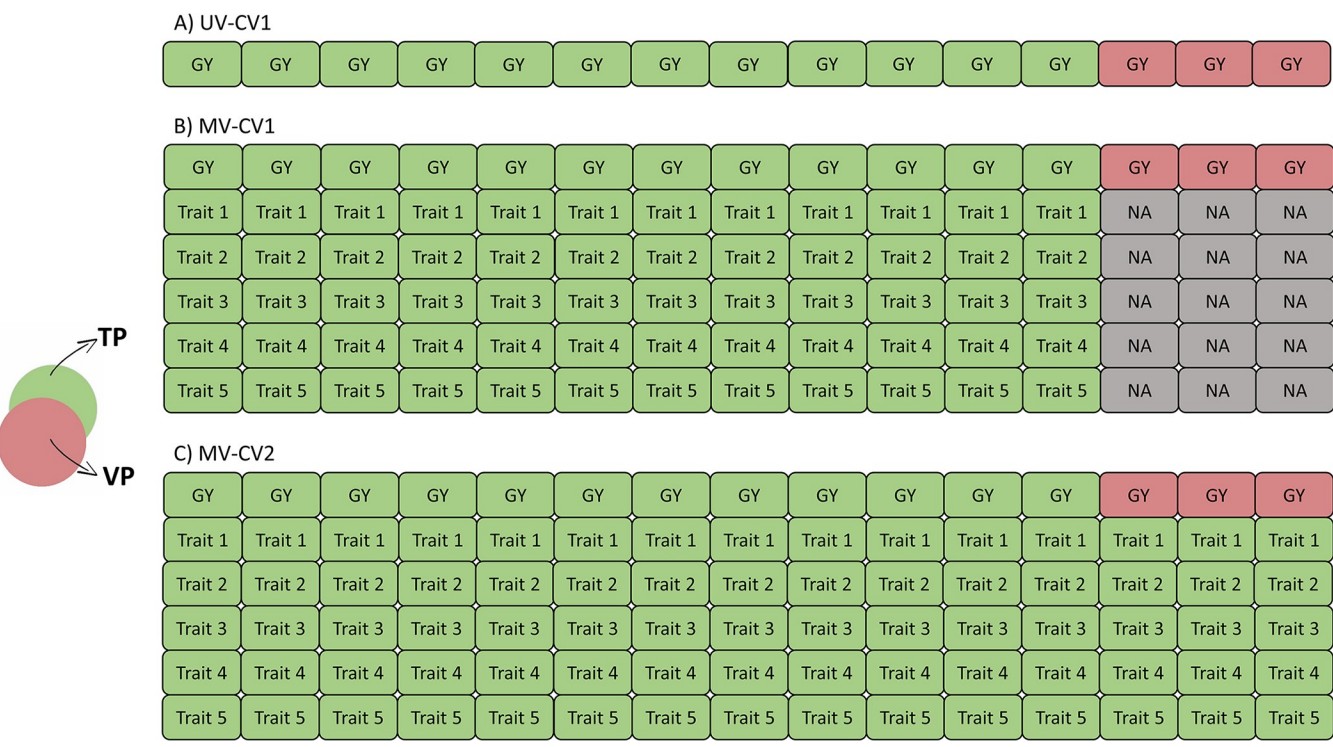

**Fig 2. Cross-validation scheme:** *Grain yield with correlated traits.* The training population (TP) is represented in green, and the validation population (VP) in red. **(A)** univariate cross-validation 1 (UV-CV1): this scheme was applied for grain yield, with 4/5 of the population used for training the model and 1/5 for validation. **(B)** Multivariate cross-validation 1 (MV-CV1): phenotypic information from the most five correlated traits were included in the model for the individuals involved in the TP. **(C)** Multivariate cross-validation 2 (MV-CV2): phenotypic information from the most five correlated traits were included in the model for the individuals involved in the TP and VP.

## Results

### General statistics, heritability, and genetic correlations

Basic summary statistics of the E-BLUEs for each trait and index under both field conditions is displayed in Table 1. Standard error was very similar between the same trait in HNW and LNR. The coefficient of variation (CV), calculated using the E-BLUE values, ranged from 0.03 (TW in LNR) to 0.16 (GY in HNW) for yield and related traits, from 0.06 (GPC in HNW and LNR) to 0.73 (GPD in LNR) for quality traits, and from 0.04 (DTM in LNR) to 0.13 (DTHD in HNW) for phenological traits. A wide range of heritability ($H^2$) for yield-related traits was observed, with values ranging from less than 0.05 for NDVI in HNW to 0.97 for TKW in LNR. Broad-sense heritability for traits related to quality ranged from 0.36 for YI to 0.83 for GPC, both evaluated in HNW. Moderate to high $H^2$ for phenology-related traits were detected, ranging from 0.48 (DTM in LNR) to 0.84 (DTHD in HNW). According to the analysis of variance (ANOVA), the factor genotype was found to be a significant source of variation in the trials for all traits (S1 Table). The field management source of variation was also statistically significant for all traits except for GY and GPD, probably due to the large amount of precipitation which confounded the effect of management. Specifically, total precipitation for the crop season 2017/2018 was 442.96 mm while the 2018/2019 crop season received 527.34 mm.

Fig 3 displays the genetic correlations between GY and all the other traits under two different field conditions. In both field conditions, positive correlations were observed between GY and other yield-related traits, while negative or weak correlations were noted with traits related

**Table 1. E-BLUEs summary statistics for all traits and indices.**

| Trait | Field Condition | Mean | Min | Max | SE | CV | $H^2$ |
|---|---|---|---|---|---|---|---|
| GY (t/ha) | HNW | 5.94 | 3.74 | 8.76 | 0.06 | 0.16 | 0.32 |
| | LNR | 5.90 | 3.55 | 8.30 | 0.05 | 0.13 | 0.53 |
| TKW (g) | HNW | 45.10 | 34.14 | 54.07 | 0.24 | 0.08 | 0.95 |
| | LNR | 49.33 | 38.69 | 59.56 | 0.25 | 0.08 | 0.97 |
| TW (g) | HNW | 76.82 | 67.01 | 82.98 | 0.19 | 0.04 | 0.78 |
| | LNR | 80.26 | 72.81 | 83.87 | 0.14 | 0.03 | 0.83 |
| GYD (index) | HNW | -7.59 | -11.80 | -4.84 | 0.06 | 0.12 | 0.20 |
| | LNR | -8.24 | -10.55 | -5.87 | 0.05 | 0.10 | 0.45 |
| NDVI (index) | HNW | 0.56 | 0.43 | 0.66 | 0.00 | 0.09 | <0.05 |
| | LNR | 0.36 | 0.24 | 0.49 | 0.00 | 0.13 | 0.22 |
| GPC (%SS) | HNW | 17.23 | 15.10 | 20.60 | 0.06 | 0.06 | 0.83 |
| | LNR | 15.23 | 13.34 | 18.11 | 0.06 | 0.06 | 0.68 |
| GPD (index) | HNW | -0.21 | -2.13 | 2.84 | 0.05 | 0.70 | 0.80 |
| | LNR | -0.15 | -1.88 | 2.65 | 0.05 | 0.73 | 0.64 |
| YI (index) | HNW | 20.01 | 11.84 | 25.40 | 0.14 | 0.11 | 0.36 |
| | LNR | 21.07 | 15.16 | 27.53 | 0.14 | 0.10 | 0.48 |
| FLA (days) | HNW | 21.68 | 16.25 | 30.75 | 0.16 | 0.12 | 0.63 |
| | LNR | 18.98 | 12.25 | 25.75 | 0.12 | 0.10 | 0.52 |
| DTHD (days) | HNW | 31.62 | 22.25 | 38.47 | 0.26 | 0.13 | 0.84 |
| | LNR | 27.73 | 19.75 | 35.00 | 0.20 | 0.11 | 0.64 |
| DTA (days) | HNW | 37.22 | 29.00 | 44.75 | 0.25 | 0.11 | 0.77 |
| | LNR | 34.49 | 27.00 | 42.50 | 0.18 | 0.08 | 0.57 |
| DTM (days) | HNW | 76.62 | 69.00 | 83.75 | 0.22 | 0.05 | 0.64 |
| | LNR | 68.87 | 62.25 | 80.75 | 0.17 | 0.04 | 0.48 |

The table includes the mean, minimum value (min), maximum value (max), standard error (SE), coefficient of variation (CV), and broad-sense heritability estimates ($H^2$). GY, grain yield; TKW, thousand kernel weight; TW, test weight; GYD, grain yield deviation; NDVI, normalized difference vegetation index; GPC, grain protein content; GPD, grain protein deviation; YI, yellow index; FLA, flag leaf appearance, DTHD, days to heading; DTA, days to anthesis; DTM days to maturity; HNW, high nitrogen and well-watered; LNR, low nitrogen and under rainfed.

to quality and phenology. For each field condition, the five traits most genetically correlated to GY were selected for inclusion in multivariate genomic prediction models. Excluding derivative indices (GYD and GPD), GY in HNW was most strongly correlated with GPC (-0.43), FLA (-0.11), YI (0.30), TW (0.32), and TKW (0.34). Under the LNR management condition, it was observed that GPC (-0.37), TKW (0.15), TW (0.20), NDVI (0.28), and YI (0.29) were the traits most strongly correlated with GY.

## Univariate genomic prediction

All samples were genotyped using an Axiom array of 420K SNP markers. After filtering, the number of markers was reduced to 63,407. Estimates of genomic prediction accuracy for all traits in both field management conditions are reported in Table 2. For yield-related traits, PA estimates ranged from 0.43 (NDVI) to 0.74 (TW) in HNW, and from 0.34 (NDVI) to 0.73 (TW) in LNR. In particular, GY showed a moderate prediction accuracy in both field managements: 0.58 in HNW and 0.53 in LNR. For quality-related traits, PA ranged from 0.44 (YI) to 0.63 (GPC) in HNW, and from 0.48 (YI) to 0.54 (GPC) in LNR. For traits related to plant phenology, moderate to high prediction accuracies were observed for both field conditions, ranging from 0.62 to 0.70 for FLA and DTHD, respectively, in HNW. Prediction accuracy

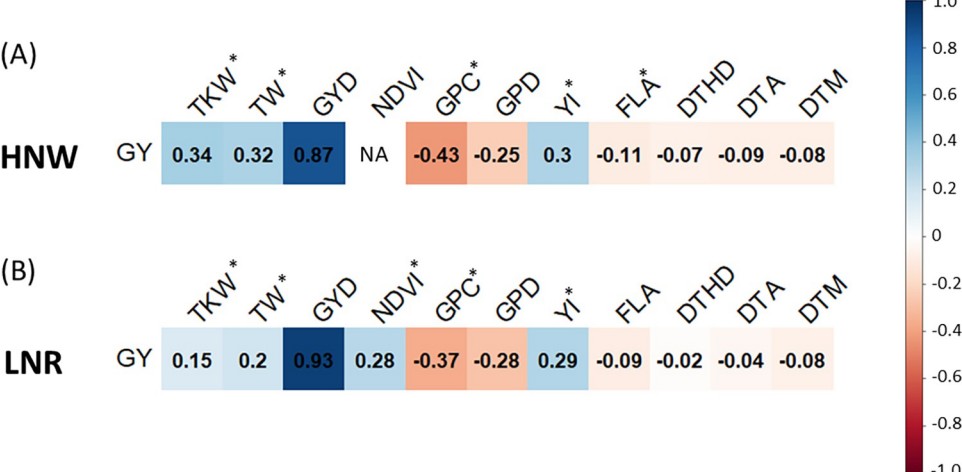

**Fig 3. Genetic correlation between grain yield and other traits.** In this plot grain yield (GY) was genetically correlated with all the other traits under evaluation and derivate indices in both **(A)** high nitrogen and well-watered (HNW) and **(B)** low nitrogen and rainfed (LNR) conditions. The five most correlated traits to grain yield were marked with the asterisk. TKW, thousand kernel weight; TW, test weight; GYD, grain yield deviation; NDVI, normalized difference vegetation index; GPC, grain protein content; GPD, grain protein deviation; YI, yellow index; FLA, flag leaf appearance, DTHD, days to heading; DTA, days to anthesis; DTM days to maturity.

estimates ranged from 0.48 (DTM) to 0.56 (DTHD) in LNR. Finally, the Fisher test was performed to discern statistical differences between the accuracy in HNW and LNR for the same trait. Prediction accuracy in HNW was significantly higher than in LNR for almost all traits (8 out 12) with the exception of TW, GYD, GPD, and YI (S1 Fig).

## Multivariate genomic prediction: Same trait in two field conditions

In this first analysis, an attempt was made to improve prediction accuracy for all traits by modelling the same traits in both HNW and LNR conditions. Estimates of PA for all traits in both field conditions, for the multivariate models, and the univariate GBLUP previously described, are displayed in Table 2. MV-CV1 was found to be ineffective for improving PA for any trait, but improvements were observed with MV-CV2. Regarding yield-related traits, significant improvements were not observed when applying the first method of cross-validation. However, significant increases in terms of PA for almost all traits were observed with MV-CV2, with variations ranging from 1.56% for NDVI in HNW using MV-GBLUP to 56.72% for the trait TKW in LNR using the RKHS model. When MV-CV1 was applied to quality traits, the largest improvement was found for GPD evaluated in HNW using RKHS (4.41%), although the difference was not statistically significant. Relevant variation in PA was detected using MV-CV2, ranging from -8.64% to 50.2% for yellow index (MV-GBLUP) and GPC (RKHS), respectively, both in LNR. Finally, when phenological traits were analyzed, a limited range of variation in prediction accuracy was found using MV-CV1, ranging from a decrease of -5.12% for DTA in the HNW condition using MV-GBLUP to an increase of 5.31% for DTM in the LNR condition using RF. By contrast, MV-CV2 significantly improved PA, with an increase of up to 23.57% for DTHD in LNR using the RKHS model.

## Multivariate genomic prediction: Grain yield with correlated traits

The goal of the second analysis was to test multivariate strategy for GY by training the models with GY alongside the five most genetically correlated traits. Consistent with the findings in

**Table 2. Prediction accuracy in univariate and multivariate using the *same trait in two field conditions* (Cond.).**

| Traits | Cond. | UV-CV1 UV-GBLUP | MV-CV1 MV-GBLUP | BBR | RKHS | SpikeSlab | RF | MV-CV2 MV-GBLUP | BRR | RKHS | SpikeSlab |
|---|---|---|---|---|---|---|---|---|---|---|---|
| GY | HNW | $0.58 \pm 0.02^{c}$ | $0.57 \pm 0.02^{c}$ | $0.58 \pm 0.02^{c}$ | $0.57 \pm 0.02^{c}$ | $0.58 \pm 0.02^{c}$ | $0.57 \pm 0.02^{c}$ | $0.67 \pm 0.02^{ab}$ | $0.69 \pm 0.02^{a}$ | $0.70 \pm 0.02^{a}$ | $0.66 \pm 0.02^{b}$ |
| | LNR | $0.53 \pm 0.02^{c}$ | $0.52 \pm 0.02^{c}$ | $0.55 \pm 0.02^{c}$ | $0.54 \pm 0.02^{c}$ | $0.53 \pm 0.02^{c}$ | $0.55 \pm 0.02^{c}$ | $0.67 \pm 0.02^{a}$ | $0.67 \pm 0.02^{a}$ | $0.67 \pm 0.02^{a}$ | $0.62 \pm 0.02^{b}$ |
| TKW | HNW | $0.57 \pm 0.03^{cd}$ | $0.55 \pm 0.02^{d}$ | $0.59 \pm 0.02^{c}$ | $0.58 \pm 0.02^{cd}$ | $0.56 \pm 0.02^{cd}$ | $0.55 \pm 0.03^{cd}$ | $0.73 \pm 0.03^{b}$ | $0.82 \pm 0.01^{a}$ | $0.83 \pm 0.01^{a}$ | $0.72 \pm 0.03^{b}$ |
| | LNR | $0.52 \pm 0.02^{cd}$ | $0.47 \pm 0.05^{d}$ | $0.51 \pm 0.03^{cd}$ | $0.53 \pm 0.02^{c}$ | $0.49 \pm 0.03^{cd}$ | $0.49 \pm 0.03^{cd}$ | $0.69 \pm 0.06^{b}$ | $0.81 \pm 0.01^{a}$ | $0.81 \pm 0.01^{a}$ | $0.71 \pm 0.02^{b}$ |
| TW | HNW | $0.74 \pm 0.01^{de}$ | $0.72 \pm 0.01^{e}$ | $0.74 \pm 0.01^{de}$ | $0.75 \pm 0.02^{d}$ | $0.74 \pm 0.01^{de}$ | $0.73 \pm 0.01^{de}$ | $0.85 \pm 0.02^{c}$ | $0.89 \pm 0.01^{ab}$ | $0.90 \pm 0.01^{a}$ | $0.87 \pm 0.01^{bc}$ |
| | LNR | $0.73 \pm 0.01^{c}$ | $0.72 \pm 0.01^{c}$ | $0.75 \pm 0.01^{c}$ | $0.75 \pm 0.01^{c}$ | $0.74 \pm 0.01^{c}$ | $0.73 \pm 0.02^{c}$ | $0.84 \pm 0.05^{b}$ | $0.89 \pm 0.01^{a}$ | $0.90 \pm 0.01^{a}$ | $0.88 \pm 0.01^{a}$ |
| GYD | HNW | $0.51 \pm 0.02^{cd}$ | $0.47 \pm 0.04^{d}$ | $0.52 \pm 0.02^{bc}$ | $0.52 \pm 0.02^{bc}$ | $0.51 \pm 0.02^{cd}$ | $0.52 \pm 0.03^{bc}$ | $0.56 \pm 0.03^{ab}$ | $0.60 \pm 0.02^{a}$ | $0.60 \pm 0.03^{a}$ | $0.56 \pm 0.03^{ab}$ |
| | LNR | $0.51 \pm 0.02^{de}$ | $0.49 \pm 0.02^{e}$ | $0.51 \pm 0.02^{de}$ | $0.53 \pm 0.02^{cd}$ | $0.52 \pm 0.02^{cde}$ | $0.52 \pm 0.02^{cde}$ | $0.58 \pm 0.04^{ab}$ | $0.59 \pm 0.02^{ab}$ | $0.60 \pm 0.02^{a}$ | $0.56 \pm 0.03^{bc}$ |
| NDVI | HNW | $0.43 \pm 0.03^{c}$ | $0.43 \pm 0.02^{c}$ | $0.44 \pm 0.02^{bc}$ | $0.44 \pm 0.03^{bc}$ | $0.42 \pm 0.02^{c}$ | $0.44 \pm 0.02^{bc}$ | $0.43 \pm 0.03^{bc}$ | $0.47 \pm 0.02^{ab}$ | $0.50 \pm 0.03^{a}$ | $0.46 \pm 0.02^{abc}$ |
| | LNR | $0.34 \pm 0.02^{c}$ | $0.38 \pm 0.03^{abc}$ | $0.36 \pm 0.02^{bc}$ | $0.34 \pm 0.03^{c}$ | $0.33 \pm 0.03^{c}$ | $0.37 \pm 0.03^{abc}$ | $0.42 \pm 0.03^{a}$ | $0.40 \pm 0.03^{ab}$ | $0.41 \pm 0.03^{a}$ | $0.38 \pm 0.02^{abc}$ |
| GPC | HNW | $0.63 \pm 0.02^{c}$ | $0.63 \pm 0.02^{c}$ | $0.63 \pm 0.02^{c}$ | $0.63 \pm 0.02^{c}$ | $0.62 \pm 0.02^{c}$ | $0.56 \pm 0.03^{d}$ | $0.79 \pm 0.01^{b}$ | $0.83 \pm 0.01^{a}$ | $0.84 \pm 0.01^{a}$ | $0.79 \pm 0.02^{b}$ |
| | LNR | $0.54 \pm 0.02^{c}$ | $0.51 \pm 0.02^{c}$ | $0.52 \pm 0.02^{c}$ | $0.53 \pm 0.03^{c}$ | $0.50 \pm 0.03^{c}$ | $0.44 \pm 0.03^{d}$ | $0.77 \pm 0.02^{ab}$ | $0.79 \pm 0.02^{a}$ | $0.81 \pm 0.01^{a}$ | $0.75 \pm 0.02^{b}$ |
| GPD | HNW | $0.56 \pm 0.03^{cd}$ | $0.57 \pm 0.02^{cd}$ | $0.57 \pm 0.03^{cd}$ | $0.58 \pm 0.02^{c}$ | $0.55 \pm 0.02^{de}$ | $0.52 \pm 0.03^{e}$ | $0.78 \pm 0.01^{a}$ | $0.79 \pm 0.01^{a}$ | $0.79 \pm 0.01^{a}$ | $0.74 \pm 0.02^{b}$ |
| | LNR | $0.52 \pm 0.02^{c}$ | $0.52 \pm 0.02^{c}$ | $0.53 \pm 0.03^{c}$ | $0.52 \pm 0.03^{c}$ | $0.48 \pm 0.03^{d}$ | $0.45 \pm 0.03^{d}$ | $0.77 \pm 0.02^{a}$ | $0.77 \pm 0.02^{a}$ | $0.78 \pm 0.01^{a}$ | $0.72 \pm 0.02^{b}$ |
| YI | HNW | $0.44 \pm 0.03^{b}$ | $0.34 \pm 0.06^{c}$ | $0.45 \pm 0.02^{b}$ | $0.44 \pm 0.03^{b}$ | $0.44 \pm 0.03^{b}$ | $0.46 \pm 0.03^{b}$ | $0.44 \pm 0.07^{b}$ | $0.59 \pm 0.02^{a}$ | $0.60 \pm 0.02^{a}$ | $0.54 \pm 0.02^{a}$ |
| | LNR | $0.48 \pm 0.02^{b}$ | $0.30 \pm 0.06^{c}$ | $0.47 \pm 0.03^{b}$ | $0.47 \pm 0.03^{b}$ | $0.48 \pm 0.02^{b}$ | $0.48 \pm 0.02^{b}$ | $0.44 \pm 0.08^{b}$ | $0.61 \pm 0.02^{a}$ | $0.62 \pm 0.02^{a}$ | $0.57 \pm 0.02^{a}$ |
| FLA | HNW | $0.62 \pm 0.01^{c}$ | $0.60 \pm 0.02^{c}$ | $0.62 \pm 0.02^{c}$ | $0.63 \pm 0.02^{c}$ | $0.63 \pm 0.02^{c}$ | $0.61 \pm 0.02^{c}$ | $0.62 \pm 0.02^{c}$ | $0.71 \pm 0.02^{a}$ | $0.71 \pm 0.02^{a}$ | $0.67 \pm 0.02^{b}$ |
| | LNR | $0.53 \pm 0.02^{cd}$ | $0.51 \pm 0.02^{d}$ | $0.55 \pm 0.02^{c}$ | $0.54 \pm 0.02^{cd}$ | $0.53 \pm 0.02^{cd}$ | $0.53 \pm 0.03^{cd}$ | $0.53 \pm 0.03^{cd}$ | $0.65 \pm 0.02^{a}$ | $0.65 \pm 0.02^{a}$ | $0.60 \pm 0.02^{b}$ |
| DTHD | HNW | $0.70 \pm 0.02^{c}$ | $0.67 \pm 0.02^{cd}$ | $0.69 \pm 0.02^{cd}$ | $0.68 \pm 0.02^{cd}$ | $0.69 \pm 0.01^{cd}$ | $0.66 \pm 0.02^{d}$ | $0.75 \pm 0.02^{ab}$ | $0.77 \pm 0.01^{a}$ | $0.77 \pm 0.01^{a}$ | $0.73 \pm 0.02^{b}$ |
| | LNR | $0.56 \pm 0.03^{d}$ | $0.55 \pm 0.04^{d}$ | $0.57 \pm 0.02^{d}$ | $0.57 \pm 0.02^{cd}$ | $0.57 \pm 0.03^{cd}$ | $0.58 \pm 0.02^{cd}$ | $0.62 \pm 0.05^{bc}$ | $0.68 \pm 0.02^{a}$ | $0.68 \pm 0.02^{a}$ | $0.63 \pm 0.02^{b}$ |
| DTA | HNW | $0.69 \pm 0.02^{b}$ | $0.65 \pm 0.05^{c}$ | $0.69 \pm 0.02^{b}$ | $0.68 \pm 0.01^{bc}$ | $0.69 \pm 0.02^{b}$ | $0.66 \pm 0.02^{bc}$ | $0.75 \pm 0.01^{a}$ | $0.75 \pm 0.01^{a}$ | $0.76 \pm 0.01^{a}$ | $0.74 \pm 0.01^{a}$ |
| | LNR | $0.52 \pm 0.03^{c}$ | $0.51 \pm 0.03^{c}$ | $0.53 \pm 0.02^{c}$ | $0.53 \pm 0.02^{c}$ | $0.53 \pm 0.02^{c}$ | $0.53 \pm 0.03^{c}$ | $0.59 \pm 0.03^{b}$ | $0.63 \pm 0.02^{a}$ | $0.64 \pm 0.02^{a}$ | $0.59 \pm 0.02^{b}$ |
| DTM | HNW | $0.67 \pm 0.02^{c}$ | $0.67 \pm 0.02^{c}$ | $0.68 \pm 0.02^{bc}$ | $0.67 \pm 0.03^{c}$ | $0.67 \pm 0.02^{c}$ | $0.65 \pm 0.02^{c}$ | $0.72 \pm 0.01^{a}$ | $0.70 \pm 0.02^{ab}$ | $0.72 \pm 0.01^{a}$ | $0.70 \pm 0.02^{ab}$ |
| | LNR | $0.48 \pm 0.03^{d}$ | $0.50 \pm 0.02^{cd}$ | $0.49 \pm 0.03^{cd}$ | $0.48 \pm 0.03^{d}$ | $0.48 \pm 0.03^{d}$ | $0.51 \pm 0.03^{cd}$ | $0.55 \pm 0.03^{ab}$ | $0.56 \pm 0.03^{ab}$ | $0.58 \pm 0.03^{a}$ | $0.53 \pm 0.02^{bc}$ |

Prediction accuracy (mean ± standard error) was observed using one univariate model as reference (UV-GBLUP) and five multivariate models such as MV-GBLUP, BRR, RKHS, Spike and slab regression, and Random Forest (RF). The multivariate model was carried out in two cross-validation schemes (MV-CV1 and MV-CV2) except for the model RF which was performed in MV-CV1 only. Duncan's test was carried on for all output within each single row of the table. GY, grain yield; TKW, thousand kernel weight; TW, test weight; GYD, grain yield deviation; NDVI, normalized difference vegetation index; GPC, grain protein content; GPD, grain protein deviation; YI, yellow index; FLA, flag leaf appearance, DTHD, days to heading; DTA, days to anthesis; DTM days to maturity; HNW, high nitrogen and well-watered; LNR, low nitrogen and under rainfed.

the first analysis, no improvements were found when MV-CV1 was applied. However, MV-CV2 increased prediction accuracy under both field conditions across most models (Fig 4). Using MV-CV1 in HNW, prediction accuracy remained roughly constant except for the MV-GBLUP model, which resulted in a significant decrease in PA of -7.34%. In contrast, a general improvement in PA, by up to ~18%, was observed when the multivariate RKHS model was applied using MV-CV2 in the same field management. For the LNR condition, a similar trend in prediction accuracy variation was observed. Indeed, no evident variation in PA was observed using MV-CV1, but statistically significant improvements were observed when MV-CV2 was applied across all models, except for MV-GBLUP (Fig 4).

## Discussion

This comprehensive study explored both univariate and multivariate genomic prediction strategies to improve the accuracy of predicting key agronomic traits in durum wheat. A key aspect involved modelling simultaneously the same trait under different field conditions using

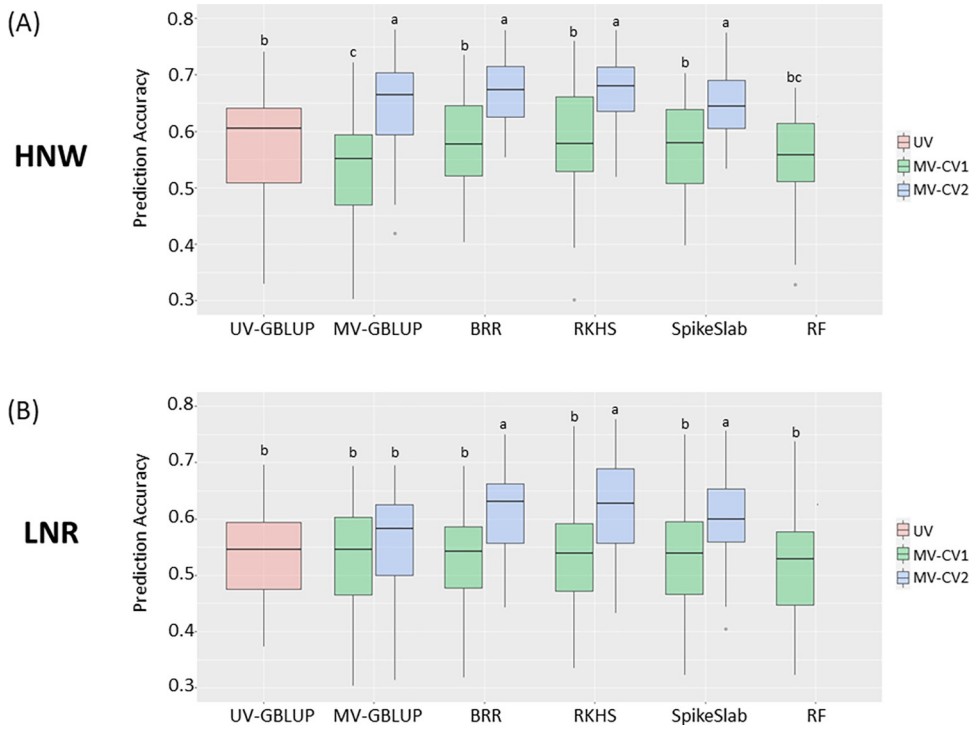

**Fig 4. Prediction accuracy for grain yield with correlated traits.** Comparison of prediction accuracy (PA) for grain yield among univariate (GBLUP) and the multivariate models (MV-GBLUP, BRR, RKHS, Spike and Slab regression, and RF) in both multivariate cross-validation schemes (MV-CV1 and MV-CV1). The Random Forest (RF) model was performed only in MV-CV1. This analysis was performed separately for (A) high nitrogen and well-watered (HNW), and (B) low nitrogen and rainfed (LNR) conditions. Duncan's test, represented by letters, was used to identify and discern statistical differences among the PAs.

multivariate genomic prediction models, systematically applied to all studied traits. Additionally, the five traits with the highest genetic correlations with grain yield were identified and incorporated into the multivariate models alongside grain yield to enhance prediction accuracy. This analysis was conducted separately for high nitrogen and well-watered (HNW) and low nitrogen and rainfed (LNR) conditions.

## Univariate genomic prediction

For yield-related traits, univariate genomic prediction exhibited variable prediction accuracy, ranging from 0.34 to 0.74. GY showed accuracies of 0.58 and 0.53 for HNW and LNR conditions respectively. As a complex quantitative trait, grain yield has been well-studied in genomic prediction [9]. Halie et al. [14] found prediction accuracy for GY using GBLUP was higher than 0.70 in a double haploid population and around 0.5 in a breeding population, this last outcome is close to our results using the same model. Genomic prediction is suitable and effective also for less complex traits such as those related to quality. Grain protein content presented 0.63 and 0.54 prediction accuracy in HNW and LNR conditions respectively, values comparable to those reported by Halie and colleagues [14]. In cereal crops, phenology-related traits are considered to have a relatively simple genetic architecture, being influenced by major genes involved in vernalization, photoperiod sensitivity, and earliness per se [49]. A higher PA in HNW compared to LNR was observed for most of the traits. This discrepancy is likely attributed to the environmental uncertainties introduced by the LNR condition, as well as the

challenges in phenotyping complex traits under such conditions. This aligns with Rabieyan et al. [50], who investigated genomic prediction accuracy for various traits in an Iranian wheat population under optimal and rainfed environments. They also reported higher prediction accuracy for several traits, including GY and TKW, in the optimal environment compared to the rainfed one. Complex traits are under the genetic control of many genes with small effects, whereas simple traits are controlled by fewer genes with large effects [51]. In our study, especially under HNW condition, traits such as grain yield exhibited a lower prediction accuracy compared to quality and phenology-related traits. This can be attributed to the prominent influence of heritability, which serves as the primary driving factor that substantially impacts prediction accuracy, as underscored by Watson et al. [31]. Specifically, quality and phenology traits had higher heritability than grain yield, which contributed to their higher prediction accuracy.

## Multivariate genomic prediction

Getting a high prediction accuracy is a fundamental prerequisite for implementing effective genomic prediction in plant breeding programs [26]. The multivariate approach has a positive impact on GP by increasing prediction accuracy compared to the univariate models, primarily due to the exploitation of the variance-covariance matrix among traits. The MV genomic prediction is particularly effective when low heritability traits are modelled together with high heritability ones [30]. Here, the prediction accuracy of 10 traits and two derivate indices was evaluated in a genomic prediction scheme including in the models the same traits in two different field conditions, followed by modelling grain yield together with the five traits that most genetically correlated with yield. In this work, the goal was to investigate whether the implementation of multivariate models could improve PA compared to a univariate model (UV-GBLUP) in durum wheat. To accomplish this aim, multivariate genomic prediction was performed using several models in two cross-validation schemes: MV-CV1, where no phenotypic information was available for the validation population, and MV-CV2, which included supplementary phenotypic information for both the training and validation populations.

We failed to improve prediction accuracy using MV-CV1 in both analyses that were performed. Contrarily, MV-CV2 consistently proved effective in increasing PA compared to the single-trait model, suggesting that including phenotypic information of correlated traits of the validation population in the training process significantly improve prediction ability. In this study, the application of MV-CV2 revealed critical improvements, with recorded values exceeding 50%. Similarly, many others studies have reported no improvements when CV1 was applied [8, 52, 53]. Gill and colleagues [54] carried out multi-trait genomic prediction modelling for various agronomic traits (yield, protein content, plant height, and heading date) from different environments in advanced lines of winter wheat. They observed PA of multi-trait CV1 was similar to that of the single-trait model for most of the trait-environment combinations. Jia and Jannink [30] concluded that the advantage of implementing CV1 is minimal compared to the single traits approach, but suggested that this multi-trait scheme may be more useful when applied to a primary trait with low heritability ($H^2 < 0.20$). This was encountered for NDVI in LNR which exhibited a low $H^2$ (0.22). Despite the statistically insignificant nature of the improvement, there was a noticeable enhancement in predictive accuracy, reaching up to 12.45%, when employing MV-CV1 with the MV-GBLUP model. However, this result did not occur for the same traits in HNW condition where an increase in PA of only up to 3.39% was detected using RKHS. Shahi et al. [55] evaluated the possibility of improving the prediction accuracy of five complex primary traits (harvest index, grain yield, grain number, spike partitioning index, and fruiting efficiency) by incorporating in the model two physiological

traits (canopy temperature, and NDVI) in 236 of soft wheat elite lines; the authors recognized that the MV-CV2 model improved predictive ability by 4.8% to 138.5% compared to single traits. The authors concluded that multi-trait genomic selection could accelerate breeding cycles and improve genetic gain for complex traits in wheat and other crops. In addition, similar results were described by other researchers for many species such as wheat [56], sorghum [57], rice [58], perennial ryegrass [53], and barley [59]. In multivariate GP, the CV2 scheme is particularly useful when the primary trait presents a low heritability, and the secondary one has high heritability [60]. This is consistent with our findings in the first analysis where phenological traits in LNR showed lower heritability than the same traits in HNW. Therefore, phenological traits in LNR demonstrated greater improvement in PA (up to 23.57%) when they were modelled with the same traits in HNW using the MV-CV2 scheme compared to the reverse operation (up to 13.04%).

In the second analysis of this work, a multivariate GP scheme was performed for grain yield incorporating the five most genetically correlated traits in the model. Similar to the initial analysis, no significant improvement was observed between univariate and multivariate CV1 for all models under either HNW and LNR conditions. On the contrary, notable increases were detected in applying the second cross-validation scheme by up to 17.97% and 17.89% in HNW and LNR respectively. Similarly, implementing grain yield together with grain protein content in multivariate models increased PA compared to the single-trait analysis in a breeding panel of durum wheat [14]. By contrast, Montesinos-López et al. [18] carried out a genomic prediction analysis for grain yield in durum wheat observing univariate GBLUP showed similar results to the multi-trait deep learning model. Our results align with those Gill et al. [54] and Shahi et al. [55], who modelled grain yield alongside agronomic and physiological traits respectively in bread wheat. Interestingly, also Semagn and colleagues [61] modelled GY together with other six agronomic traits such as plant height, TKW, DTHD, DTM, TW, and GPC comparing uni- and multivariate models in three spring wheat populations. The authors used 75% and 25% of the population to train and to test the single-trait model, subsequently, they applied two multi-trait schemes: MT1, where all traits were included in the models for the training set, and no information from the validation set individuals were used; and MT2, where the model assumed that the validation set was observed for some traits but not others. They concluded that MT2 was superior compared to the single-trait and MT1 models increasing PA by an overall average of 52.8%. Finally, these last outcomes suggest the effectiveness of multivariate CV2 in improving prediction accuracy.

Prediction accuracy is influenced by many factors such as sample size, the relationship between training and validation population, marker density, and statistical models [62]. In our study, several multivariate models were performed such as MV-GBLUP, BRR, RKHS, Spike-Slab, and RF. Overall, it was recognized the supremacy in PA of the BRR and RKHS models over MV-GBLUP, SpikeSlab, and RF, among yield- quality- and phenology-related traits. For instance, BRR outperformed MV-GBLUP for TKW, and RKHS outperformed MV-GBLUP for TW under MV-CV1 in HNW. Similarly, the models BRR and RKHS showed a prediction accuracy statistically higher than MV-GBLUP and SpikeSlab for traits FLA, DTHD, and DTM using MV-CV2 in LNR. This outcome could be explained by the non-linear nature of some models (such as RKHS), that probably be able to capture non-additive effects [63]. Zingaretti et al. [24] carried out a comparison of genomic prediction models among Bayesian LASSO (BL), BRR, Bayesian ridge regression general model (BRR-GM), RKHS, and a deep learning model in polyploid outcrossing species using a conventional single trait scheme. They found that BRR-GM produced the best result for fruit weight prediction, BL, BRR, and RKHS showed the highest prediction accuracy for early marketable yield, and RKHS and BRR-GM were the best models for total marketable weight. The authors reported that the RKHS model

was able to capture complex interaction patterns, as previously explained by Gianola et al. in 2006 and 2008 [64, 65]. In a multivariate genomic prediction study, Sandhu et al. [66] compared several different single- and multi-traits models such as GBLUP, Bayesian models, and machine- and deep-learning models predicting grain yield and protein content in a dataset of 650 recombinant inbred lines of wheat. They observed multi- outperformed single-trait by up to 28.50%, and they observed that RF and multi-layer perception were the best-performing models for both traits. As observed by Zingaretti et al. [24] regarding RKHS models, similar findings were reported by Sandhu et al. [66] for machine and deep learning models. This study affirmed that machine and deep learning models possessed a remarkable level of flexibility in mapping complex interactions between predictors and responses. Consequently, this flexibility enables these models to effectively interpret the trends observed within the current dataset. Multivariate GBLUP was used to assess prediction accuracy for grain yield, incorporating high-throughput phenotyping traits as secondary traits in a CV2 approach. Consistent with our findings, the authors have demonstrated the clear superiority of the multivariate approach over the univariate strategy [36]. Here, an increase in PA was observed when using MV-CV2 instead of MV-CV1, while GY was modelled along with the five most genetically related traits within the field condition. The latter result is very promising because it highlighted the potential of multivariate analysis to significantly improve genetic gain for GY by adding other agronomic traits in genomic prediction models. While MV-CV1 is a preferred option for its potential savings in phenotyping-related costs and time, MV-CV2 demonstrates the effectiveness of incorporating additional field conditions into genomic selection-based breeding programs as well as including correlated traits for GY prediction, leading to improved prediction accuracy and enhanced genetic gain. Modelling different correlated traits alongside grain yield is especially valuable in predicting pre-yield trials in durum wheat breeding programs. Indeed, generations such as F4 in durum wheat breeding are constrained by small seed quantities, making it challenging for breeders to conduct yield trials. However, traits like TKW, TW, NDVI and others are readily detectable in these trials. Therefore, training genomic prediction models using data from the previous year for both grain yield and its correlated traits, along with traits observed in pre-yield trial (target population), offers an effective strategy to enhance prediction accuracy for grain yield. This approach empowers breeders to make more accurate selections or discards based on predictions, starting from pre-yield trial before lines are subjected to yield testing. As climate change poses unprecedented challenges, the adaptability of our approach to marginal areas, such as drought-prone environments, becomes particularly appealing. The ability to make more accurate selections based on predictions, even in the pre-yield trials of the breeding pipeline, holds the key to developing resilient genotypes capable of thriving in diverse conditions.

## Conclusion

This investigation underscores the potential of employing a multivariate genomic prediction strategy to elevate the accuracy of predicting grain yield and various agronomic traits in durum wheat. The multivariate approach consistently outperformed the univariate counterpart, particularly when incorporating information from secondary traits in the model, as evidenced by the results from multivariate cross-validation 2. This cross-validation technique showed notable improvements in prediction accuracy across various scenarios: encompassing the same trait in different field conditions and modelling grain yield alongside the five most genetically correlated traits. The implications of this study might represent practical insights that could be applied in durum wheat breeding programs that rely on genomic prediction. The prospect of leveraging information gleaned from secondary traits, especially in predicting

grain yield during pre-yield trials, emerges as a highly promising avenue. As highlighted in our study, this strategic application not only enhances prediction accuracy but, more significantly, could improve genetic gain in durum wheat breeding programs. These findings present a compelling case for the integration of multivariate genomic prediction methodologies, paving the way for the development of more resilient and productive durum wheat cultivars in the future.

## Supporting information

**S1 Table. Analysis of variance (ANOVA).** Raw data were modelled to discern statistical differences among all the factors under examination such as genotype, year, condition, block, genotype by year, genotype by condition, and genotype by year by condition. GY, grain yield; TKW, thousand kernel weight; TW, test weight; GYD, grain yield deviation; NDVI, normalized difference vegetation index; GPC, grain protein content; GPD, grain protein deviation; YI, yellow index; FLA, flag leaf appearance, DTHD, days to heading; DTA, days to anthesis; DTM days to maturity; Gen, Genotype; Cond, Condition. Levels of significance:, $<0.001$ (***), $<0.01$ (**), $<0.05$ (*), non-significant (ns).
(DOCX)

**S1 Fig. Univariate genomic prediction accuracy for all traits.** UV-GBLUP model was performed for all traits in both high nitrogen and well-watered (HNW, pink), and (B) low nitrogen and under rainfed (LNR, teal) conditions. The Fisher test was performed to discern statistical differences between the accuracies of two field conditions for the same traits. The statistical significance was indicated by the asterisk. GY, grain yield; TKW, thousand kernel weight; TW, test weight; GYD, grain yield deviation; NDVI, normalized difference vegetation index; GPC, grain protein content; GPD, grain protein deviation; YI, yellow index; FLA, flag leaf appearance, DTHD, days to heading; DTA, days to anthesis; DTM days to maturity.
(TIF)

## Acknowledgments

We express our gratitude to AL and to the University of Minnesota for hosting PV during a sabbatical period of five months. We wish to acknowledge Giuseppe Petruzzino for his support with the bioinformatics operations. We extend our gratitude also to Vito De Gregorio, Aniello Padalino, Alfredo Morcone, Antonio Bruno, and Antonio Gallo for their exceptional technical support during the field trials conducted at CREA-CI.

## Author Contributions

**Conceptualization:** Aaron Lorenz, Pasquale De Vita, Nicola Pecchioni.

**Data curation:** Paolo Vitale, Giovanni Laidò, Ivano Pecorella.

**Formal analysis:** Paolo Vitale, Gabriella Dono.

**Investigation:** Pasquale De Vita, Nicola Pecchioni.

**Methodology:** Paolo Vitale, Ivano Pecorella.

**Supervision:** Vishnu Ramasubramanian, Aaron Lorenz, Pasquale De Vita, Nicola Pecchioni.

**Validation:** Paolo Vitale.

**Visualization:** Paolo Vitale.

**Writing – original draft:** Paolo Vitale.

**Writing – review & editing:** Paolo Vitale, Vishnu Ramasubramanian, Aaron Lorenz, Pasquale De Vita, Nicola Pecchioni.

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
