## [Decision Letter · Decision Letter 0]

17 Nov 2023

PONE-D-23-28445Multivariate genomic prediction for agronomic traits in durum wheat under two field conditionsPLOS ONE

Dear Dr. Vitale,

Thank you for submitting your manuscript to PLOS ONE. After careful consideration, we feel that it has merit but does not fully meet PLOS ONE’s publication criteria as it currently stands. Therefore, we invite you to submit a revised version of the manuscript that addresses the points raised during the review process.

Please revise the manuscript taking into consideration the reviewers' comments.  Please submit your revised manuscript by Jan 01 2024 11:59PM. If you will need more time than this to complete your revisions, please reply to this message or contact the journal office at plosone@plos.org. Please include the following items when submitting your revised manuscript:A rebuttal letter that responds to each point raised by the academic editor and reviewer(s). You should upload this letter as a separate file labeled 'Response to Reviewers'.A marked-up copy of your manuscript that highlights changes made to the original version. You should upload this as a separate file labeled 'Revised Manuscript with Track Changes'.An unmarked version of your revised paper without tracked changes. You should upload this as a separate file labeled 'Manuscript'.If applicable, we recommend that you deposit your laboratory protocols in protocols.io to enhance the reproducibility of your results. Protocols.io assigns your protocol its own identifier (DOI) so that it can be cited independently in the future. For instructions see: https://journals.plos.org/plosone/s/submission-guidelines#loc-laboratory-protocols. Additionally, PLOS ONE offers an option for publishing peer-reviewed Lab Protocol articles, which describe protocols hosted on protocols.io. Read more information on sharing protocols at https://plos.org/protocols?utm_medium=editorial-email&utm_source=authorletters&utm_campaign=protocols.

We look forward to receiving your revised manuscript.

Kind regards,

Sindhu Sareen

Academic Editor

PLOS ONE

4. Please include a caption for figure 4.

5. We are unable to open your Supporting Information file [Supporting_information.rar]. Please kindly revise as necessary and re-upload.

Reviewers' comments:

Reviewer's Responses to Questions

**Comments to the Author**

1. Is the manuscript technically sound, and do the data support the conclusions?

Reviewer #1: Yes

Reviewer #2: Yes

2. Has the statistical analysis been performed appropriately and rigorously? 

Reviewer #1: Yes

Reviewer #2: Yes

3. Have the authors made all data underlying the findings in their manuscript fully available?

Reviewer #1: No

Reviewer #2: Yes

4. Is the manuscript presented in an intelligible fashion and written in standard English?

Reviewer #1: Yes

Reviewer #2: No

5. Review Comments to the Author

Reviewer #1: The manuscript by Vitale and coauthors presents an evaluation of a set of genomic prediction models in durum wheat. The manuscript is well written, with well-documented methods, and findings might be useful for the durum wheat breeding community. However, my primary concern lies in the extent of the new information presented in the study as it has been limited to mere model comparison using a training population. It will be great if authors can validate the multivariate models in an independent breeding population, if possible.

Comments/suggestions for authors to improve the manuscript:

- The usage of terms ‘HIGH’ and ‘LOW’ in reference to the two growing regimes is quite confusing, especially when presenting results for model comparisons. I would suggest the authors replace these terms with another appropriate words.

-I would request the authors to provide the data along with the R script available so that breeders can actually make use of these findings.

- The authors estimated the genetic correlation between traits in META-R using equation given in Lines 161-62. Why did authors not use the genomic data (markers’ matrix) for estimating the genetic correlations? I would suggest accounting the markers information for estimating genetic correlation among traits.

- Though the methods are well documented, the section presenting methodology for different cross validation schemes lacks clarity. For instance, the authors used the multivariate GP model to predict grain yield using ‘only’ highly correlated traits. Nevertheless, the authors should mention these traits in methods as well as while presenting results.

- In continuation of above, the authors used five correlated traits to predict grain yield. If I am correct, this also included test weight and TKW as covariate in the MT model. In practicality, it would not make sense to use a post-harvest trait to predict grain yield. However, it might be useful in specific scenarios where the data from previous years may be included in models to predict grain yield. The authors need to discuss these scenarios in detail and present the discussion in a way to elucidate how breeders can make use of these models in their programs.

- I would suggest including recent references reporting the evaluation of multivariate GP in wheat.

- The authors used ‘GP pathways’ several times in the manuscript. The word ‘pathway(s)’ does not seem to fit in this context.

- Line 20: I believe GP has not been implemented for many years, but more been evaluated.

- Line 325: statically?

There are few typos in the manuscript, and I would request the authors to run

Reviewer #2: Generally, this is an interesting paper and the description of the study mostly understandable and logical. However, the paper has to be improved with major modifications. Also, serious consideration should be given to the English grammar. Although it is understandable, it must be improved.

Comments and suggestions for authors

Title should be modified to “Univariate and Multivariate genomic prediction for agronomic traits in durum wheat under two field conditions”.

Moreover, the manuscript has some deficiencies in its citation practice, particularly in the introduction. The authors need to properly align their statements with relevant and appropriately cited literature, it undermines the credibility and integrity of the work.

The over-use of the pronoun "We” must be avoided in whole paper.

Line 53: delete "is currently becoming" to "becomes"

Line 57: replace "genomic selection" by "GS"

Line 67: Therefore, genomic selection appears ….

Line 69: citation is missing.

Line 78 and line 80: repetition

Line 94: delete "With this in our mind"

Line 115-Line 116: on November 28th and December 7th

Line 117: for the first- and second-year trial….

Line 217: please correct the explanation of formulae according to Gill et al (2023)

Gill, H. S., Brar, N., Halder, J., Hall, C., Seabourn, B. W., Chen, Y. R., St. Amand, P., Bernardo, A., Bai, G., Glover, K., Turnipseed, B., & Sehgal, S. K. (2023). Multi-trait genomic selection improves the prediction accuracy of end-use quality traits in hard winter wheat. The Plant Genome, 00, e20331. https://doi.org/10.1002/tpg2.20331

Table1: please delete unit column and add trait unit in trait column

Table2: you should add this part “GY, grain yield; TKW, thousand kernel weight; TW, test weight; GYD, grain yield deviation; NDVI, normalized difference vegetation index; GPC, grain protein content; GPD, grain protein deviation; YI, yellow index; FLA, flag leaf appearance, DTHD, days to heading; DTA, days to anthesis; DTM days to maturity.” in table title

Delete RF column for MV-CV2

Discussion part needs deep improvement: the discussion of results is pretty generic.

Line 436: Halie et al.

Line 420 to Line 426: this part should be placed in introduction part.

Line 439 to 442: we cannot compare genomic prediction in maize with that in durum wheat

Line 449: quality and phenology traits exhibited….

Although Conclusions is optional, it should be added in your paper to highlight the paper findings.

In Supplementary Table 1, you should add 3-way interaction "Genotype*Year*Condition"

6. PLOS authors have the option to publish the peer review history of their article (what does this mean?). If published, this will include your full peer review and any attached files.

Reviewer #1: No

Reviewer #2: No

---

## [Author Response · Author response to Decision Letter 0]

26 Apr 2024

“Journal requirements:

https://journals.plos.org/plosone/s/file?id=ba62/PLOSOne_formatting_sample_title_authors_affiliations.pdf”

All the figures and the supporting information are named as requested by the journal.

“2. We note that the grant information you provided in the ‘Funding Information’ and ‘Financial Disclosure’ sections do not match.

When you resubmit, please ensure that you provide the correct grant numbers for the awards you received for your study in the ‘Funding Information’ section.”

We apologize for the oversight. The accurate "funding information" is provided in the attached document, and it has also been appropriately included in the manuscript. “The research was partially funded by European Commission, Grant Agreement number: 727247 — SolACE — H2020-SFS-2016-2017/H2020-SFS-2016-2, and by Italian Ministry of Agriculture, Food and Forestry Policies (MiPAAF), sub-project ‘Tecnologie digitali integrate per il rafforzamento sostenibile di produzioni e trasformazioni agroalimentari (AgroFiliere)’ (AgriDigit programme) (DM 36503.7305.2018 of 20/12/2018).”

“3. We note that you have indicated that data from this study are available upon request. PLOS only allows data to be available upon request if there are legal or ethical restrictions on sharing data publicly. For more information on unacceptable data access restrictions, please see http://journals.plos.org/plosone/s/data-availability#loc-unacceptable-data-access-restrictions.

We will update your Data Availability statement on your behalf to reflect the information you provide.”

Molecular markers were uploaded in the public repository figshare (https://doi.org/10.6084/m9.figshare.22674667.v1). Similarly, the phenotypic dataset can be obtained by downloading from the public repository Figshare (10.6084/m9.figshare.25554993). 

“4. Please include a caption for figure 4.”

The caption of figure 4 was added to the manuscript.

“5. We are unable to open your Supporting Information file [Supporting_information.rar]. Please kindly revise as necessary and re-upload.”

We have regenerated the zip file, now including S1_Fig and S1_Table. We trust that this adjustment will not cause any inconvenience.

“6. Please review your reference list to ensure that it is complete and correct. If you have cited papers that have been retracted, please include the rationale for doing so in the manuscript text, or remove these references and replace them with relevant current references. Any changes to the reference list should be mentioned in the rebuttal letter that accompanies your revised manuscript. If you need to cite a retracted article, indicate the article’s retracted status in the References list and also include a citation and full reference for the retraction notice.”

We changed the reference list in accordance with the new suggestions provided by the reviewers. And any changes is under track mode. 

“Reviewer #1: The manuscript by Vitale and coauthors presents an evaluation of a set of genomic prediction models in durum wheat. The manuscript is well written, with well-documented methods, and findings might be useful for the durum wheat breeding community. However, my primary concern lies in the extent of the new information presented in the study as it has been limited to mere model comparison using a training population. It will be great if authors can validate the multivariate models in an independent breeding population, if possible.”

Thank you for your thoughtful feedback. We appreciate your positive remarks regarding the clarity of our manuscript and the documentation of our methods. Regarding your concern about the limited scope of new information, we acknowledge the importance of independent validation in genomic prediction studies. However, we would like to emphasize that cross-validation remains a robust and widely accepted strategy for model testing in wheat breeding. While we understand the value of an independent breeding population for validation, unfortunately, we do not currently have access to another population suitable for this type of validation. Nevertheless, we are committed to refining and extending our study in future works.

“Comments/suggestions for authors to improve the manuscript:

- The usage of terms ‘HIGH’ and ‘LOW’ in reference to the two growing regimes is quite confusing, especially when presenting results for model comparisons. I would suggest the authors replace these terms with another appropriate words.”

We appreciate the reviewer's feedback and acknowledge the confusion caused by the terms 'HIGH' and 'LOW' in our manuscript. To address this concern, we have revised the acronyms for the two field conditions. We now use 'HNW' for high nitrogen and well-watered condition, and 'LNR' for low nitrogen and under rainfed condition.

“-I would request the authors to provide the data along with the R script available so that breeders can actually make use of these findings.”

We recognize the significance of sharing R scripts for the benefit of other researchers and breeders. However, it is important to note that no new scripts were generated for this paper; instead, we utilized functions provided by the packages referenced in the manuscript. To facilitate easy code retrieval for readers, we have incorporated hyperlinks and references to relevant publications (Lines 225-226, 250-251, 265-266) where the scripts employed in this study can be found.

“- The authors estimated the genetic correlation between traits in META-R using equation given in Lines 161-62. Why did authors not use the genomic data (markers’ matrix) for estimating the genetic correlations? I would suggest accounting the markers information for estimating genetic correlation among traits.”

We appreciate the reviewer's insightful comment. While META-R is a robust tool for estimating genetic correlations, we recognize the merit in incorporating molecular marker information for a comprehensive analysis. In response to this suggestion, we have calculated genetic correlations using the markers' information, as detailed in lines 181-189 of the manuscript.

“- Though the methods are well documented, the section presenting methodology for different cross validation schemes lacks clarity. For instance, the authors used the multivariate GP model to predict grain yield using ‘only’ highly correlated traits. Nevertheless, the authors should mention these traits in methods as well as while presenting results.”

We concur with the reviewer's observation regarding the lack of clarity in the "Cross-validation schemes" paragraph. In response, we have made substantial revisions to enhance the clarity of this section in both sub-paragraphs: “Same trait in two field conditions”, and “Grain yield with correlated traits”. Finally, we also added the most genetically correlated traits for both field conditions.

“- In continuation of above, the authors used five correlated traits to predict grain yield. If I am correct, this also included test weight and TKW as covariate in the MT model. In practicality, it would not make sense to use a post-harvest trait to predict grain yield. However, it might be useful in specific scenarios where the data from previous years may be included in models to predict grain yield. The authors need to discuss these scenarios in detail and present the discussion in a way to elucidate how breeders can make use of these models in their programs.”

We appreciate the insightful feedback provided by reviewer #1. The highlighted point is indeed valid, and we acknowledge the significance of this approach, especially in pre-yield trials. In the manuscript, specifically in lines 564-576, we have duly emphasized the importance of this strategy, noting that traits like TKW, TW, and others are observable in early-stage trials, where grain yield might not be readily observed. 

“- I would suggest including recent references reporting the evaluation of multivariate GP in wheat.”

We have enriched the manuscript by incorporating additional references pertinent to multivariate GP in wheat (Lines 90-98).

“- The authors used ‘GP pathways’ several times in the manuscript. The word ‘pathway(s)’ does not seem to fit in this context.”

The word “pathway(s)” was removed from the manuscript.

“- Line 20: I believe GP has not been implemented for many years, but more been evaluated.”

We changed it according to the reviewer’s suggestion. 

“- Line 325: statically?”

We meant statistically; the error was corrected.

“There are few typos in the manuscript, and I would request the authors to run”

The manuscript was reviewed for addressing typos issue. 

“Reviewer #2: Generally, this is an interesting paper and the description of the study mostly understandable and logical. However, the paper has to be improved with major modifications. Also, serious consideration should be given to the English grammar. Although it is understandable, it must be improved.”

We appreciate the positive feedback provided by reviewer #2 on the overall interest and logical coherence of our manuscript. Additionally, we have carefully reviewed the paper, giving due attention to the English grammar concerns highlighted by reviewer #2, and made necessary improvements to enhance the overall quality of the manuscript.

“Comments and suggestions for authors

Title should be modified to “Univariate and Multivariate genomic prediction for agronomic traits in durum wheat under two field conditions”.

Moreover, the manuscript has some deficiencies in its citation practice, particularly in the introduction. The authors need to properly align their statements with relevant and appropriately cited literature, it undermines the credibility and integrity of the work.

The over-use of the pronoun "We” must be avoided in whole paper.”

We appreciate the valuable feedback from the reviewer, and we have implemented the suggested changes to enhance the manuscript. Specifically, the title and introduction have been revised as per the reviewer's recommendations. In response to the concern about the overuse of the pronoun "we," substantial efforts have been made to minimize its usage throughout the paper, aiming for greater clarity and conciseness.

Line 53: delete "is currently becoming" to "becomes" corrected 

Line 57: replace "genomic selection" by "GS" corrected

Line 67: Therefore, genomic selection appears …. corrected

Line 69: citation is missing. added

Line 78 and line 80: repetition removed

Line 94: delete "With this in our mind" removed

Line 115-Line 116: on November 28th and December 7th corrected

Line 117: for the first- and second-year trial…. Corrected

“Line 217: please correct the explanation of formulae according to Gill et al (2023)

Gill, H. S., Brar, N., Halder, J., Hall, C., Seabourn, B. W., Chen, Y. R., St. Amand, P., Bernardo, A., Bai, G., Glover, K., Turnipseed, B., & Sehgal, S. K. (2023). Multi-trait genomic selection improves the prediction accuracy of end-use quality traits in hard winter wheat. The Plant Genome, 00, e20331. https://doi.org/10.1002/tpg2.20331” We have corrected the explanation of the elements according to the reviewer’s suggestions and according to Sandhu et al., 2022 https://doi.org/10.3389/fgene.2022.831020

Table1: please delete unit column and add trait unit in trait column. corrected

Table2: you should add this part “GY, grain yield; TKW, thousand kernel weight; TW, test weight; GYD, grain yield deviation; NDVI, normalized difference vegetation index; GPC, grain protein content; GPD, grain protein deviation; YI, yellow index; FLA, flag leaf appearance, DTHD, days to heading; DTA, days to anthesis; DTM days to maturity.” in table title. added

Delete RF column for MV-CV2. deleted 

“Discussion part needs deep improvement: the discussion of results is pretty generic.”

We appreciate the constructive feedback from the reviewer regarding the Discussion section. In response to this valuable input, we made comprehensive improvements to delve deeper into the discussion of results. Specifically, we have expanded our exploration of multivariate genomic prediction applications in the durum wheat breeding program, as also suggested by reviewer #1. 

Line 436: Halie et al. corrected

Line 420 to Line 426: this part should be placed in introduction part. The part mentioned by reviewer #2 was correctly replaced in the introduction section. 

Line 439 to 442: we cannot compare genomic prediction in maize with that in durum wheat. corrected

Line 449: quality and phenology traits exhibited…. corrected

“Although Conclusions is optional, it should be added in your paper to highlight the paper findings.”

The conclusion was added as suggested by reviewer #2

“In Supplementary Table 1, you should add 3-way interaction "Genotype*Year*Condition"

The 3-way interaction "Genotype*Year*Condition" was added to table S1.

---

## [Decision Letter · Decision Letter 1]

20 Aug 2024

PONE-D-23-28445R1Univariate and multivariate genomic prediction for agronomic traits in durum wheat under two field conditionsPLOS ONE

Dear Dr. Vitale,

Thank you for submitting your manuscript to PLOS ONE. After careful consideration, we feel that it has merit but does not fully meet PLOS ONE’s publication criteria as it currently stands. Therefore, we invite you to submit a revised version of the manuscript that addresses the points raised during the review process.

We look forward to receiving your revised manuscript.

Kind regards,

Sindhu Sareen

Academic Editor

PLOS ONE

Journal Requirements:

Reviewers' comments:

Reviewer's Responses to Questions

**Comments to the Author**

1. If the authors have adequately addressed your comments raised in a previous round of review and you feel that this manuscript is now acceptable for publication, you may indicate that here to bypass the “Comments to the Author” section, enter your conflict of interest statement in the “Confidential to Editor” section, and submit your "Accept" recommendation.

Reviewer #1: (No Response)

Reviewer #2: All comments have been addressed

2. Is the manuscript technically sound, and do the data support the conclusions?

Reviewer #1: Yes

Reviewer #2: Yes

3. Has the statistical analysis been performed appropriately and rigorously? 

Reviewer #1: Yes

Reviewer #2: Yes

4. Have the authors made all data underlying the findings in their manuscript fully available?

Reviewer #1: No

Reviewer #2: Yes

5. Is the manuscript presented in an intelligible fashion and written in standard English?

Reviewer #1: No

Reviewer #2: No

6. Review Comments to the Author

Reviewer #1: The authors have revised the manuscript and addressed several of my concerns.

I have a few comments for the authors:

1. About the codes, I am unsure if there would be any issue in creating an online repository and loading the scripts used for analysis. This exercise is important to allow the reproducibility of the results when all other data is being made available. Secondly, the authors provided the links to BGLR and SOMMER repositories. Please note that these resources merely provide information on fitting these models but no information to cross-validate them while using MTGP. I again request the Authors to create a Github (or another repository) and load the codes for usability/reproducibility.

2. I still do not see any references to 'recent literature' on multivariate prediction in wheat when there are a lot of recent publications. I believe it authors can use that information to present how breeding programs are using multivariate prediction for a variety of traits in different scenarios.

3. In the updated version, authors have used a matrix notation to represent the GP models. Please make sure the equations are presented appropriately. In this notation, include all terms in bold except the overall mean (u). The same procedure would need to be done for all terms mentioned in the text/paragraph, except for the overall mean and the variance components of the genetic and residual effects.

4. Please provide the parameters that were used to fit the models and cross-validations. For instance, the burn-ins, iteration, independent runs, etc. Again, providing a code makes it much easier for the reader to go through the modeling method at once.

5. There are several typos in the text and please check the manuscript carefully. For example, GPLUP instead of GBLUP, MN-CV instead of MV-CV, etc.

Reviewer #2: The manuscript is good in its present form and accepted with minor English issues that authors should go accordingly.

Thus, English grammar revision is well recommended for the full manuscript.

Indeed in the discussion part: the first paragraph should be paraphrased (L571 -> L578).

The use of pronoun "We" should be avoided.

7. PLOS authors have the option to publish the peer review history of their article (what does this mean?). If published, this will include your full peer review and any attached files.

Reviewer #1: No

Reviewer #2: **Yes: **Amir Souissi

---

## [Author Response · Author response to Decision Letter 1]

7 Sep 2024

Dear Sindhu Sareen,

We would like to thank you and the reviewers for your valuable time and insightful comments on our manuscript titled “Univariate and multivariate genomic prediction for agronomic traits in durum wheat under two field conditions.” We appreciate the constructive feedback, which has helped us improve the quality and clarity of our work. In this response letter, we have carefully addressed each of the reviewers’ comments, and we have made the necessary revisions to the manuscript accordingly.

Below, you will find a detailed point-by-point response to the reviewers’ suggestions, along with explanations of the changes made. We believe these revisions have strengthened the manuscript and hope it now meets the expectations of both the reviewers and the journal.

Thank you once again for the opportunity to revise our work, and we look forward to your feedback.

Sincerely,

Paolo Vitale

Reviewer #1: 

Comment 1. About the codes, I am unsure if there would be any issue in creating an online repository and loading the scripts used for analysis. This exercise is important to allow the reproducibility of the results when all other data is being made available. Secondly, the authors provided the links to BGLR and SOMMER repositories. Please note that these resources merely provide information on fitting these models but no information to cross-validate them while using MTGP. I again request the Authors to create a Github (or another repository) and load the codes for usability/reproducibility.

Answer 1: As recommended by the reviewer #1, a GitHub repository containing all the necessary code for running multivariate genomic prediction has been shared. (https://github.com/paolovitale777/Code-univariate-and-multivariate-genomic-prediction/blob/main/Code). In the text it was reported in lines 280-281

Comment 2: I still do not see any references to 'recent literature' on multivariate prediction in wheat when there are a lot of recent publications. I believe it authors can use that information to present how breeding programs are using multivariate prediction for a variety of traits in different scenarios.

Answer 2: Thank to reviewer #1 for highlighting this, additional recent examples of multivariate genomic prediction have been added to the introduction (lines 90 to 111) and the discussion (lines 501-502, 556-559).

Comment 3: In the updated version, authors have used a matrix notation to represent the GP models. Please make sure the equations are presented appropriately. In this notation, include all terms in bold except the overall mean (u). The same procedure would need to be done for all terms mentioned in the text/paragraph, except for the overall mean and the variance components of the genetic and residual effects.

Answer 3: In response to Reviewer 1's comment, we have followed the convention of using bold typeface for vectors and matrices, while leaving scalar values in regular type. According to this convention, the overall mean and variance components are not bold, as noted by the reviewer. Please consider that overall means "µ" ^"'" in equations 8 and 10 represents a vector and for this reason, this term is in bold. If we have overlooked any other terms, please let us know and we will address them promptly.

Comment 4: Please provide the parameters that were used to fit the models and cross-validations. For instance, the burn-ins, iteration, independent runs, etc. Again, providing a code makes it much easier for the reader to go through the modeling method at once.

Answer 4: We are grateful to the reviewer for pointing this out. In response, we have provided the code where all this information can be accessed (see answer 1).

Comment 5: There are several typos in the text and please check the manuscript carefully. For example, GPLUP instead of GBLUP, MN-CV instead of MV-CV, etc.

Answer 5: We carefully reviewed the manuscript to address any typos that might be present in the manuscript. 

Reviewer #2:

Thus, English grammar revision is well recommended for the full manuscript.

Comment 1: Indeed in the discussion part: the first paragraph should be paraphrased (L571 -> L578).

Answer 1: We are grateful the reviewer #2 for highlighting this. Indeed, we improved the clarity and the readability of the first paragraph of the discussion. 

Comment 2: The use of pronoun "We" should be avoided.

Answer 2: The use of the pronoun "We" has been drastically reduced.

---

## [Editor Report · Decision Letter 2]

9 Sep 2024

Univariate and multivariate genomic prediction for agronomic traits in durum wheat under two field conditions

PONE-D-23-28445R2

Dear Dr. Vitale,

We’re pleased to inform you that your manuscript has been judged scientifically suitable for publication and will be formally accepted for publication once it meets all outstanding technical requirements.

Kind regards,

Sindhu Sareen

Academic Editor

PLOS ONE
---

## [Editor Report · Acceptance letter]

3 Nov 2024

PONE-D-23-28445R2 

PLOS ONE

Dear Dr. Vitale, 

I'm pleased to inform you that your manuscript has been deemed suitable for publication in PLOS ONE. Congratulations! Your manuscript is now being handed over to our production team.

Kind regards, 

on behalf of

Dr. Sindhu Sareen 

Academic Editor

PLOS ONE